# Benchmarking Interpretability in Healthcare Using Pattern Discovery and Disentanglement

**DOI:** 10.3390/bioengineering12030308

**Published:** 2025-03-18

**Authors:** Pei-Yuan Zhou, Amane Takeuchi, Fernando Martinez-Lopez, Malikeh Ehghaghi, Andrew K. C. Wong, En-Shiun Annie Lee

**Affiliations:** 1System Design Engineering, University of Waterloo, Waterloo, ON N2L 3G1, Canada; 2Department of Computer Science, University of Toronto, Toronto, ON M5S 1A1, Canada; amane.takeuchi@mail.utoronto.ca (A.T.); malikeh.ehghaghi@mail.utoronto.ca (M.E.); annie.lee@ontariotechu.ca (E.-S.A.L.); 3Computer and Information Sciences Department, Fordham University, Bronx, NY 10023, USA; fmartinezlopez@fordham.edu; 4Faculty of Science, Ontario Tech University, Oshawa, ON L1G 0C5, Canada

**Keywords:** interpretability, electronic health records, pattern discovery, pattern disentanglement, clinical notes

## Abstract

The healthcare industry seeks to integrate AI into clinical applications, yet understanding AI decision making remains a challenge for healthcare practitioners as these systems often function as black boxes. Our work benchmarks the Pattern Discovery and Disentanglement (PDD) system’s unsupervised learning algorithm, which provides interpretable outputs and clustering results from clinical notes to aid decision making. Using the MIMIC-IV dataset, we process free-text clinical notes and ICD-9 codes with Term Frequency-Inverse Document Frequency and Topic Modeling. The PDD algorithm discretizes numerical features into event-based features, discovers association patterns from a disentangled statistical feature value association space, and clusters clinical records. The output is an interpretable knowledge base linking knowledge, patterns, and data to support decision making. Despite being unsupervised, PDD demonstrated performance comparable to supervised deep learning models, validating its clustering ability and knowledge representation. We benchmark interpretability techniques—Feature Permutation, Gradient SHAP, and Integrated Gradients—on the best-performing models (in terms of F1, ROC AUC, balanced accuracy, etc.), evaluating these based on sufficiency, comprehensiveness, and sensitivity metrics. Our findings highlight the limitations of feature importance ranking and post hoc analysis for clinical diagnosis. Meanwhile, PDD’s global interpretability effectively compensates for these issues, helping healthcare practitioners understand the decision-making process and providing suggestive clusters of diseases to assist their diagnosis.

## 1. Introduction

While Artificial Intelligence (AI) has numerous applications, from surgical aids to clinical diagnostics, AI’s potential to revolutionize disease detection and patient categorization is remarkable [1,2,3,4]. Utilizing deep learning models has proven effective and efficient in detecting signs of disease. For instance, it is from 50 to 70 % more accurate and 50,000 times faster than a radiologist working alone [3]. However, this technological advancement presents challenges. The primary concern lies in the inherent opacity of AI models, often referred to as the ‘black box’ due to their lack of interpretability [5]. This opacity necessitates a trade-off between model accuracy and a clearer understanding of its decision-making process [6]. While numerous state-of-the-art (SoTA) methodologies exist to clarify the decision-making rules of black-box models in AI research, most rely on post hoc analysis [7], which provides insights only after the model has been trained. 

This paper extends the study [8] by applying the Pattern Discovery and Disentanglement (PDD) [9] approach to the MIMIC-IV dataset [10] and comparing it to popular post hoc interpretability tools. PDD uses clinical notes to diagnose diseases and organize patients’ symptomatic patterns into distinct clusters, representing statistically significant trait associations. This method generates an interpretable knowledge base, aiding decision making and providing valuable clinical insights. By comparing PDD with popular post hoc interpretability techniques of black-box deep learning models, our study highlights PDD’s strengths and potential applications in enhancing the transparency and reliability of AI-driven healthcare solutions.

First, we extract features from the free text in the MIMIC-IV dataset using Term Frequency-Inverse Document Frequency (TF-IDF) and Topic Modeling (TM) techniques, creating three datasets as follows: a TM dataset with 50 features, a unigram TFIDF dataset with 250 features, and a unigram and bigram TF-IDF dataset with 250 features. We train and evaluate a suite of cutting-edge deep learning models on these three datasets. Then, we apply post hoc interpretability methods including Feature Permutation [11], Integrated Gradients [12], and Gradient SHAP (SHapley Additive exPlanations) [13] to reveal the contribution of each input feature to disease classification on the best-performing deep learning model. Finally, we evaluate the interpretability results from post hoc analysis using the *sufficiency* [14], *comprehensiveness* [14], and *sensitivity* [15] metrics. Applying PDD to the same TFIDF and TM datasets, we cluster disease codes and generate an interpretable knowledge base. Comparing interpretability insights from post hoc analysis and the PDD outputs, we highlight PDD’s advantage in enhancing AI transparency in healthcare. While post hoc analysis has limitations, like local interpretability and limited feature importance ordering, our study demonstrates how PDD and its knowledge base address these issues (Figure 1). 

## 2. Related Work 

Ravì et al. [16] emphasized that while deep learning models offer diverse applications in health informatics, such as medical imaging, and public health, they present significant limitations. Our literature review explored the state of AI and deep learning models in healthcare analysis and the importance of model interpretability in this field.

### 2.1. AI in Healthcare 

Horng et al. [17] trained a machine learning (ML) model to effectively classify sepsis using textual data and NLP techniques, achieving a test AUC (Area Under the ROC curve) score of 0.85. Additionally, the AI application extends to transformers, which serve as the backbone of large language models (LLMs), such as BERT [18]. Domain-specific variants of BERT, such as BioBERT and Clinical BERT, have outperformed state-of-the-art models in many NLP tasks within biological and clinical contexts [2,19]. However, LLMs face challenges in understanding context-specific negations, transforming phrases like ‘abstinence from alcohol’ into ‘alcohol dependence syndrome’ [4]. Additionally, Johnson [20] highlighted the potential biases and errors in AI and ML models, emphasizing the need for careful training and validation. Lyell et al. [21] also pointed out that AI systems potentially have a bias against minority groups, people with disabilities, and individuals with mental health problems.

### 2.2. Model Interpretability and Its Importance

NLP models are increasingly applied in various fields including healthcare, law, and commerce [10], which prompts us to explore the definition of model interpretability and its necessity in these fields. Doshi-Velez and Kim [22] described interpretability in the context of machine learning as the ability to make terms understandable to humans. However, they also argued that interpretability is not necessary when (i) there are no serious consequences for unacceptable results or (ii) the problem is adequately studied and validated in their application and we have evidence to trust the model’s decisions [22]. In contrast, Lipton [23] emphasized the importance of interpretability, using the example of machine learning models used to predict crime rates for police deployment. These models may have made accurate predictions, but they cannot account for racial biases persistent in training data, possibly over-policing some neighborhoods. Hence, establishing trust is crucial because many people may be affected by model decisions. Therefore, this motivates us to have more transparent AI models, specifically in the field of healthcare.

## 3. Proposed Model

### 3.1. Dataset

The present study utilizes the Medical Information Mart for Intensive Care (MIMIC) dataset, more specifically, MIMIC-IV and MIMIC-IV-Notes. MIMIC-IV [10] is a comprehensive compilation of de-identified electronic health records from over 40,000 ICU patients at the Beth Israel Deaconess Medical Center between 2008 and 2019 [24]. MIMIC-IV Notes comprises clinical free-text notes for the patients within the MIMIC-IV dataset [25]. The discharge summaries and diagnosed International Classification of Disease (ICD-9) codes of each admitted patient are extracted as the input and labels, respectively. Table 1 presents a comprehensive description of the ten most prevalent ICD-9 diagnostic codes. 

### 3.2. Dataset Preprocessing

Our preprocessing pipeline is inspired by a previously established framework [8,26]. Datasets were created based on the top ten most frequently diagnosed ICD-9 codes, divided into training, validation, and test groups with a 70−10−20% split. We extracted features for clean clinical notes using the TM (motivated by [17]) and Term Frequency-Inverse Document Frequency (TF-IDF) methods, creating three datasets as follows: a TM dataset with 50 features, an unigram TF-IDF dataset with 250 features, and a bigram-included TF-IDF dataset with 250 features. To perform analysis on the topic clusters on the TM dataset, we utilized pyLDAvis [27] for interactive visualization and the GPT4 [28] model for topic naming. 

### 3.3. Benchmarking Baseline Deep Learning Models

We implemented several popular deep learning models and two machine learning models to establish baselines for evaluating the performance of the PDD method proposed in this study. Namely, a Convolutional Neural Network Model (CNN) (motivated by [29]), a Gate Recurrent Unit Model (GRU) (motivated by [30]), a Long Short-Term Memory Model (LSTM) (motivated by [31]), a Recurrent Neural Network Model (RNN) (motivated by [30]), and a Multi-Filter Residual Convolutional Neural Network Model (MultiResCNN) [32] (motivated by [29]). In addition, we also compared the overall performance with two other classic machine learning algorithms as follows: Support Vector Machine (SVM) and Random Forest (RF).

### 3.4. Clustering Using Pattern Discovery and Disentanglement

PDD [9] consists of two main steps as follows: Pattern Disentanglement and Pattern Clustering [8]. Disentanglement uses Principal Component Analysis (PCA) on the statistical residual matrix (SRM) to categorize associations between Attributes Values (AVs) into a disentangled space. The projections of AVs on the principal components are grouped into AV groups with AVs exhibiting statistically significant associations with other AVs in the group. From each AV group, the AVs on the records (i.e., patients) are further clustered into AV subgroups if they are statistically associated with at least one AV in the subgroup and not with any AV outside of it. Consequently, each AVSG represents a functionally and statistically independent cluster of AVs, presumed to originate from a distinct source [8,9,33]. The PDD output is structured into three parts in the PDD knowledge base as follows: the Knowledge Section (source of knowledge); the Pattern Section (significant AV associations); and the Data Section (record IDs and linked to other sections). Records are clustered by matching them to the class with the most associated patterns, with accuracy metrics evaluating the clustering effectiveness [9,33].

### 3.5. Post Hoc Interpretability Methods for Deep Learning Models 

While the PDD model provides a detailed and interpretable knowledge base, conventional black-box models lack such inherent transparency. To bridge this gap and enhance the understandability of these models’ decision-making process, we applied various post hoc interpretability techniques, including Feature Permutation, Integrated Gradients, and Gradient SHAP. 

Feature Permutation is a permutation-based algorithm that computes the importance of features at a global level [11]. Each feature’s value is randomly shuffled while keeping other features unchanged, and then, the algorithm computes the difference between the original model outputs and the shuffled model outputs.Integrated Gradients is a multi-step algorithm designed for feature importance estimation in deep learning models [12]. The process starts with establishing a baseline input, which serves as a point of comparison against the actual input. Following this, the algorithm constructs a linear path between the baseline and actual inputs. Crucially, Integrated Gradients compute the integral of the model’s output gradients with respect to each input feature along this linear path. Lastly, an importance score is assigned to each feature.Gradient SHAP estimates the SHapley Additive exPlanations (SHAP) value by calculating the expected value of gradients after conducting random sampling from baseline distribution [13]. This algorithm starts by adding white noise to every input observation by *x* times, where *x* is the total number of samples. From a particular baseline distribution, it picks a random point on the path between the baseline and the input. Then, the gradient of model outputs with respect to the randomly picked points is calculated. The described process is iterated a number of times so that the expected value of gradients can be approximated [13].

### 3.6. Evaluation of Interpretability Approaches 

While post hoc interpretability tools reveal the importance of input features for deep learning models, their faithfulness remains to be determined. According to Jacovi and Goldberg [34], a faithful interpretation accurately represents the reasoning process behind a model’s prediction, where the reasoning process is the underlying mechanism of how the model comes to its conclusion. Ideally, we would directly compare the explanation of a model given by the interpretability technique and the actual underlying reasoning process of the predictor model. However, the exact reasoning of the predictor model is not directly accessible to the user. Thus, most methods of determining the faithfulness of an interpretability method involve comparing various aspects of the model prediction after manipulating an input.

In this study, we employed three methods to measure the faithfulness of interpretability techniques as follows: *sufficiency*, *comprehensiveness*, and *sensitivity*. According to Chan et al. [35], *comprehensiveness* and *sufficiency* outperformed other popular faithfulness criteria regarding diagnosticity (diagnosticity is a measure of how more favored faithful interpretation is over unfaithful interpretation) with lower time complexity. While *sufficiency* and *comprehensiveness* are removal-based faithfulness metrics, *sensitivity* measures the faithfulness of the interpretation technique by adding noise to a group of important features [15], giving us another aspect to evaluate the faithfulness of the model with. We define the following terms for the equations.
*m* is the model.*B* is the set, [1, 5, 10, 20, 50] as stated in the original paper.$x_{i}$ is the input observation *i.*$r_{ik}$ are the top *k%* rationales (important features) for input $x_{i}$.$r_{ik}^{*}$ are the top *k%* rationales (important features) applied to Gaussian noise for input $x_{i}$.$m{\left(x_{i}\right)}_{j}$ is the original predicted probability for class *j.*$m{\left(x_{i}\backslash r_{ik}\right)}_{j}$ is the predicted probability for class *j* after removing rationales $r_{ik}$ from $x_{i}$.


*Sufficiency* measures whether the snippets extracted from the rationale are sufficient for a model to make a prediction, and it was proposed by DeYoung et al. [14]. The input to the model is modified only to include tokens of high importance, expecting that the label of the prediction model remains the same with high confidence (1). (1)$$Sufficiency=\frac{1}{\left|B\right|}\;\sum_{k\in B}\;m{\left(x_{i}\right)}_{j}-m{\left(r_{ik}\right)}_{j}$$

*Comprehensiveness* assesses whether all the important tokens are needed to make a prediction [14]. It is measured by removing tokens identified as important instead of only keeping the important tokens. This method expects a decrease in confidence when an explanation is faithful (2).(2)$$Comprehensiveness=\frac{1}{\left|B\right|}\;\sum_{k\in B}m{\left(x_{i}\right)}_{j}-m{\left(x_{i}\backslash r_{ik}\right)}_{j}\;$$

*Sensitivity* assesses how sensitive an explanation is to noise [15]. Local but adversarial noise is added to a sequence of tokens identified as the most relevant to a prediction, and the magnitude of noise needed to result in a change in prediction is measured. A lower magnitude indicates greater faithfulness since a predictor model should be more sensitive to the perturbation of more relevant tokens when compared to the perturbation of irrelevant tokens [15]. Sensitivity is sometimes used in a context where the noise is random and not necessarily generated to be adversarial [15]. In our context, we applied Gaussian noise to the set of features identified as important. We measure the difference in probabilities of the original and altered dataset (3).(3)$$Sensitivity=\frac{1}{\left|B\right|}\;\sum_{k\in B}m{\left(x_{i}\right)}_{j}-m{\left(r_{ik}^{*}\right)}_{j}$$

## 4. Results and Discussions

In this section, the results and discussions will be presented.

### 4.1. Comparing the Performance of Baseline Classification Models

As the PDD algorithm effectively handles imbalanced data [9,33], we aimed to compare whether other classifiers can do the same. To ensure consistency in our experimental setup, we did not apply any oversampling or undersampling techniques to the deep learning or machine learning models. Additionally, handling imbalanced data is not the primary focus of this study. 

Given the class imbalance in the distribution of ICD-9 codes across the training, validation, and test sets, we report the results in both macro and micro perspectives. Instead of traditional accuracy, we use balanced accuracy, a more robust measure that accounts for class imbalances. Therefore, we evaluated models using comprehensive performance metrics, including Macro ROC-AUC, Micro ROC-AUC, Macro F1 Score, Micro F1 Score, Balanced Accuracy, Macro Precision, Micro Precision, Macro Recall, and Micro Recall, to evaluate them based on different aspects (Table 2). 

As shown in Table 2, the CNN, GRU, and MultiResCNN baseline models exhibit approximately 20% better performance using TF-IDF features than TM features. Although the overall performance of these models is 20% better in the TF-IDF datasets, the general trend is consistent across both TF-IDF datasets and TM datasets, with the MultiResCNN performing the best among other deep learning models in most metrics. The CNN model had better results for both TF-IDF datasets than the MultiResCNN on the Macro F1, Balanced Accuracy, and Macro Recall metrics. However, the differences in these metric scores are subtle, and all deep learning models have similar scores. Additionally, the performance of the machine learning models is similar to that of the deep learning models.

Meanwhile, we observed that PDD achieves the highest balanced accuracy across all datasets. It also demonstrates exceptional performance on the bigram-included TF-IDF dataset (the bigram-included TF-IDF dataset contains both unigram and bigram data to better capture the context of free text data), achieving high scores in Macro F1, Micro F1, Balanced Accuracy, Macro Precision, Micro Precision, Macro Recall, and Micro Recall. These results validate PDD’s clustering capability and highlight the effectiveness of its knowledge base representation, which is generated through its clustering algorithm. This establishes PDD as a traceable and transparent white-box model [9,33].

### 4.2. Discussions on the Interpretability of the PDD Model

The output of the PDD, illustrated in Figure 2, offers enhanced interpretability, deepening our understanding of the associations between ICD-9 codes and the feature value associations within each cluster of the ICD-9 codes. The first three columns of the knowledge base show the Knowledge Section, represented by the Disentangled Space Unit (DSU), followed by the Pattern Section, which displays the statistically significant associations among the AVs, such as tokens or topics. The remaining columns form the Data Section demonstrate the origins of the patterns and how these discovered patterns encompass patients’ records. As illustrated in Figure 2a, two contrasting groups are generated within the first disentangled space within the Knowledge Section. The first group, labeled as ‘1’ in the second column, is associated with ICD-9 codes 272.4, 272, 244.9, and 401.9, related to metabolic and cardiovascular disorders. Conversely, the second group, denoted as ‘2’ in the second column, corresponds to ICD-9 codes 70.54, 38.9, and 8.45, related to infectious and gastrointestinal diseases. Further subgroups identified in the third column are each associated with different ICD-9 codes. The Data Section shows records possessing the discovered patterns, with only the first nine records displayed in the figure due to space limitations. According to the output, records 1–3 and 5–9 possess patterns that consistently align with their original class labels. 

Figure 2b presents a similar interpretable analysis using TM features. In the Knowledge Section, within the first disentangled space, Group 1 shows patterns associated with ICD-9 codes 70.7, 70.54, 38.9, and 8.45, which may be related to infectious hepatogastroenteric conditions; and Group 2 shows patterns associated with ICD-9 codes 272.4, 272, 285.9, and 401.9, which may relate to metabolic and cardiovascular conditions.

More specific patterns are listed below.

ICD-9 codes 70.7, 70.54, 38.9, and 8.45 (infectious hepatogastroenteric conditions) are associated with *topic 4* (Orthostatic Hypotension in Elderly Patients), *topic* 5 (Neurological Examination: Focus on Cranial Nerves), *topic 9* (The Role of Psychiatry in Managing Suicidal Ideation and Behavior) and *topic 10* (Cerebral Infarcts and Aneurysms: A Focus on Vascular Neurology) in low probabilities.ICD-9 codes 272.4, 272, 285.9, and 401.9 (metabolic and cardiovascular conditions) exhibit different patterns and are associated with *topic 4* (Orthostatic Hypotension in Elderly Patients), *topic 5* (Neurological Examination: Focus on Cranial Nerves), *topic 9* (The Role of Psychiatry in Managing Suicidal Ideation and Behavior), and high probability is associated with *topic 10* (Cerebral Infarcts and Aneurysms: A Focus on Vascular Neurology) and *topic 23* (Diagnostic Approach to Gastrointestinal Malignancies) in medium probabilities.ICD-9 codes 250, 272.4, and 272 (metabolic syndrome disorders) are associated with *topic 4* (Orthostatic Hypotension in Elderly Patients), and *topic 5* (Neurological Examination: Focus on Cranial Nerves) with higher probabilities.

Additionally, in the Data Section, six of the first ten records (ID = 1,2,5,7,9,10) are correctly assigned to their classes. 

Furthermore, Figure 2c portrays interesting results from the knowledge base output with a bigram-included TF-IDF dataset. The first group identified in the table consists of ICD-9 codes 250, 272.4, 272, and 401.9, representing cardiometabolic disorders. The second group was associated with ICD-9 codes 70.7, 70.54, 38.9, and 8.45, representing infectious hepatointestinal conditions. The third group comprises ICD-9 codes 250 and 244.9, representing endocrine metabolic disorders. Consider the following for how some feature tokens are associated with each clustered group. 

ICD-9 codes 250, 272.4, 272, and 401.9 (cardiometabolic disorders) are associated with low probabilities of tokens *antibiotic*, *diarrhea*, *infection*, *initially*, *likely*, *lymph*, *setting*, *started*, and *treated*. Meanwhile, they have a high probability for tokens such as *aspirin* and *cardiac*.ICD-9 codes 70.7, 70.54, 38.9, and 8.45 (infectious hepatointestinal conditions) demonstrate the opposite trend in relation to the first group (cardiometabolic disorders). These labels are associated with high probabilities of tokens *antibiotics*, *diarrhea*, *infection*, *initially*, *likely*, *lymph*, *setting*, *started* and *treated* but low probabilities of tokens *aspirin* and *cardiac*. Notably, this group of diseases is infectious, and ICD-9 code 8.45 (intestinal induction due to clostridium difficile) often accompanies symptoms of diarrhea.ICD-9 codes 250 and 244.9 (endocrine metabolic disorders) were associated with high probabilities of tokens *antibiotics*, *diabetes*, and *infection*, while they were also associated with low probabilities of tokens *cardiac* and *started*.

In summary, compared to the results obtained from Topic Modeling, the PDD’s knowledge base output from the TF-IDF datasets provides more clearly interpretable results. More specifically, the output from the bigram-included TF-IDF dataset demonstrated an insightful interpretation. 

### 4.3. Benchmarking Post Hoc Interpretability Techniques

We observed similar sets of tokens are gathered in Feature Permutation and Gradient SHAP for complications of unspecified septicemia (Figure 3). Integrated Gradients also have *Sepsis* as the second most influential feature (Figure 3c). Additionally, the term *hypothyroidism* has relatively high SHAP and feature permutation values for predicting septicemia. *Hypothyroidism* occurs when the thyroid gland fails to produce enough thyroid hormones to meet the body’s needs. It is a condition often associated with sepsis. Therefore, these interpretability tools have assembled reasonable tokens to predict each label class.

The difference between the interpretability tools mentioned above and PDD is that while those tools can rank associations between features and targets, they do not specify how these associations occur. In contrast, PDD provides associations at the feature value level, not just at the feature level. For example, both SHAP and PDD identify the association between *tablet* and ICD-9: 38.9, but PDD can specify patterns such as *tablet* occurring with a low probability, in the range of 0 to 0.9%, associated with ICD-9 code 38.9. Additionally, PDD can trace back to show which records are covered by these patterns, illustrating not only what the patterns are but also where they originate. 

### 4.4. Evaluation of Interpretability Techniques 

Our investigation into the faithfulness and effectiveness of post hoc interpretability methods is of utmost importance in the medical discipline. A fundamental question drives this experiment as follows: How can we trust these tools to uncover the underlying mechanisms in deep learning models? Therefore, evaluating post hoc interpretability tools is a critical step in ensuring the reliability of deep learning models. To evaluate the faithfulness and effectiveness of the post hoc interpretability tools, we conducted experiments with respect to the following two aspects: (1) how well the interpretability tool gathered the important features and (2) how well the interpretability tool signifies the importance of each feature. While the first question addresses the faithfulness of the interpretability tool, the latter question was motivated by observing that the Feature Permutation and Gradient SHAP methods shared similar groups of features, but the order of importance varied within these groups. We calculated the faithfulness criteria for the first question using the formula defined in section II-F. To validate the second question, we created plots to visualize whether the interpretability tool effectively assigned an order of importance. We measured each metric by considering a window with size five and measuring the distance between the probability the model assigns to each output class in the original and altered samples. We calculated the mean distance of class probabilities for original and altered samples for the first five most important features, then the second five most important features, and so on. For sufficiency, if the interpretability technique is faithful and the order of important features is reliable, the mean distance will gradually increase while we move the window. Likewise, for comprehensiveness, altering the important features will produce more distant output probabilities from the original sample. For sensitivity, a greater value in the mean distance will indicate the interpretability tool’s effectiveness in assigning the correct order to each feature.

#### 4.4.1. Faithfulness of Interpretability Tools

Table 3 summarizes the general faithfulness criteria performed on Gradient SHAP, Feature Permutation, and Integrated Gradients on the MultiResCNN model. Consider that *comprehensiveness* and *sufficiency* are removal-based faithfulness criteria, whereas *sensitivity* adds noise to the existing dataset instead of removing parts of it. Removal-based metrics help us understand the importance of specific attributes by monitoring changes when these are excluded. In contrast, non-removal-based metrics, such as *sensitivity*, offer an additional standpoint by showing how the model’s outputs respond to subtle perturbations, which test the reliability of the interpretability methods.

Notice that the higher values for *comprehensiveness* and *sensitivity* imply a more faithful interpretation of the model. In contrast, the lower values for *sufficiency* imply a more faithful interpretation of the model. For all criteria (Unweighted Average), Feature Permutation performs the best with relatively minor differences from Gradient SHAP but more considerable differences for the Integrated Gradients method. This observation is reasonable as Gradient SHAP and Feature Permutation selected similar features. However, these results suggest that Feature Permutation is slightly more faithful than Gradient SHAP, but they perform critically better than Integrated Gradients.

#### 4.4.2. Effectiveness of Interpretability Tools

For *sufficiency* plots, we expect the values to gradually increase as the window moves to the lower feature importance, if the order of feature importance is faithful. In contrast, we expect the values to decrease progressively for *comprehensiveness* and *sensitivity* plots as the window moves to the lower feature importance, if the order feature important is faithful. For both Feature Permutation and Gradient SHAP, we see the trend that the *sufficiency* plots increase until the 2nd to 4th window, and it gradually decreases or stays constant, indicating some features are not ordered to their actual contribution to the prediction (Figure 4a). The plot for Integrated Gradients follows a similar trend but with more fluctuations, suggesting a more considerable variance in the order of feature importance.

In contrast, we observed interesting spikes in the sensitivity plots. For both Feature Permutation and Gradient SHAP, several spikes were detected in the plots, indicating that introducing noise to certain features significantly affected the output classification probabilities. For instance, in the class ‘Other and Unspecified Hyperlipidemia’. The sensitivity Feature Permutation plot has two spikes at the 5th and 6th windows, suggesting that features in these windows had a larger impact in classifying this label with respect to its neighbor features (i.e., features in the 3rd or 4th window) (Figure 4b). Similarly, Gradient SHAP sensitivity plot encloses a larger spike at the 5th window (Figure 4c). In contrast, the Integrated Gradients have more spikes, and the spikes are larger, implying features were not appropriately ordered by their importance (Figure 4d).

#### 4.4.3. Discussions

Overall, Feature Permutation and Gradient SHAP achieved similar faithful scores, both performing better than the Integrated Gradients method. However, the Feature Permutation performed slightly better than the Gradient SHAP. Based on the plots to assess the effectiveness of the interpretability tools, we observed the order of feature importance did not necessarily align with their actual contribution to the decision-making process. This is because spikes in the *sensitivity* plots convey that adding noise to some features causes a greater impact on classification probabilities than others. Altogether, our results imply that Feature Permutation and Gradient SHAP can identify informative features to some extent, and the ranking of their importance does not necessarily reflect the underlying true order of feature importance. PDD addresses the limitations of post hoc analysis by providing interpretability at the statically interdependence level, making it both precise and robust. Although direct comparison with other ML models at the feature level is challenging, PDD compensates by assigning a range of values associated with each feature rather than single importance values. This approach uncovers a more intrinsic and reliable decision-making process that is easily associated with specific clusters of disease and makes it more interpretable in clinical diagnostics. Moreover, PDD not only clusters the feature tokens for specific ICD-9 codes but also groups similar ICD-9 codes at the label level. This is particularly beneficial in medical applications as some ICD-9 codes share overlapping symptoms. For instance, diabetes is often linked to high levels of LDL cholesterol which relates to hypercholesterolemia—an association captured by PDD (Figure 2a). These label-level clustering results provide valuable insights for clinicians, aiding in more accurate diagnosis.

### 4.5. Computational Cost Discussion

Post hoc interpretability techniques, such as Gradient SHAP, Integrated Gradients, and Feature Permutation, introduce additional computational overhead to an already complex model. A fundamental limitation of these methods is that they do not inherently make models interpretable; instead, they add an extra layer of approximation to extract insights from opaque models.

Among these techniques, Gradient SHAP requires computing expectations of gradients over multiple noisy samples, making it computationally expensive, especially for deep neural networks. Integrated Gradients follows a similar approach, requiring multiple forward and backward passes to approximate the contribution of each feature. While Feature Permutation is comparatively lightweight, it still incurs additional computational costs due to repeated model evaluations to measure feature importance.

In contrast, our proposed PDD method is fundamentally different. Since it is designed to be inherently interpretable, it does not require post hoc explanations. The interpretability is embedded within the model structure itself, eliminating the need for additional computational layers to “decode” model behavior. Moreover, PDD does not rely on iterative sampling or gradient-based approximations, making it more scalable and computationally efficient compared to post hoc methods that require GPU acceleration for large-scale applications.

## 5. Conclusions

Motivated by the critical need of interpretability in AI-driven healthcare, we benchmark the Pattern Discovery and Disentanglement (PDD) algorithm, designed to process clinical notes and help diagnose diseases while generating interpretable output grids for clinical experts. Unlike conventional black-box models, PDD inherently reveals the decision-making processes, thereby enhancing transparency and trust in AI-driven predictive analysis. This intrinsic interpretability makes PDD a unique and valuable tool in clinical settings, where understanding algorithmic decisions is essential for patient care.

Our comprehensive evaluations demonstrate that PDD achieves performance comparable to existing methods, validating its effectiveness in machine learning. Furthermore, we have illustrated that conventional post hoc interpretability techniques, such as Feature Permutation, Integrated Gradients, and Gradient SHAP, have notable limitations in achieving clear and reliable insights. In contrast, PDD directly addresses these challenges by producing a structured knowledge base representation, offering transparency and robust decision-making processes. These findings reinforce the importance of integrating inherently interpretable models in clinical environments to ensure reliable and explainable AI-driven decision making. Despite these promising results, our feature extraction method, TF-IDF, has limitations in capturing longer-ordered token relations, which may affect contextual understanding. 

In summary, the Pattern Discovery and Disentanglement (PDD) algorithm marks a significant advancement in AI-driven healthcare by offering a robust, interpretable, and clinically relevant approach to pattern discovery for knowledge representation and predictive analysis. Its capacity to provide clear insights into decision-making processes enhances trust, transparency, and efficacy in AI-powered clinical diagnostics, underscoring the necessity of interpretable AI for responsible and impactful healthcare applications.

Our future work will explore several directions to enhance the effectiveness and applicability of this study. First, to address the limitations of the feature extraction method, we will incorporate higher-order n-grams to improve contextual representation in clinical text analysis. Second, we will extend our evaluation to diverse medical datasets across different clinical settings. Finally, while PDD has demonstrated its effectiveness in processing text data, we will further investigate its applicability to image data. Specifically, we aim to explore its integration with multimodal healthcare data, such as combining clinical text with imaging or structured EHR data, to enhance its clinical utility.

## Figures and Tables

**Figure 1 bioengineering-12-00308-f001:**
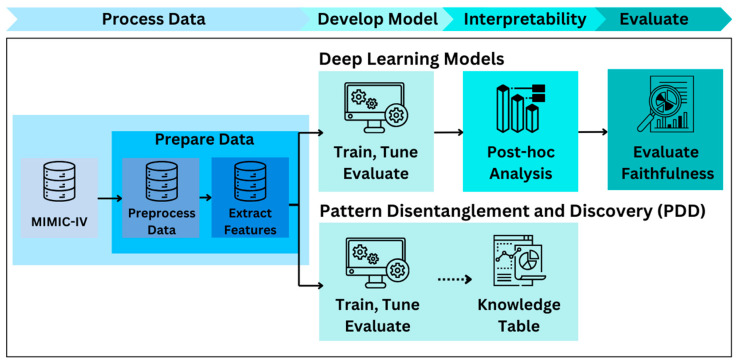
Overview of the benchmarking process. Comparison of the benchmarking models and the PDD for interpretability.

**Figure 2 bioengineering-12-00308-f002:**
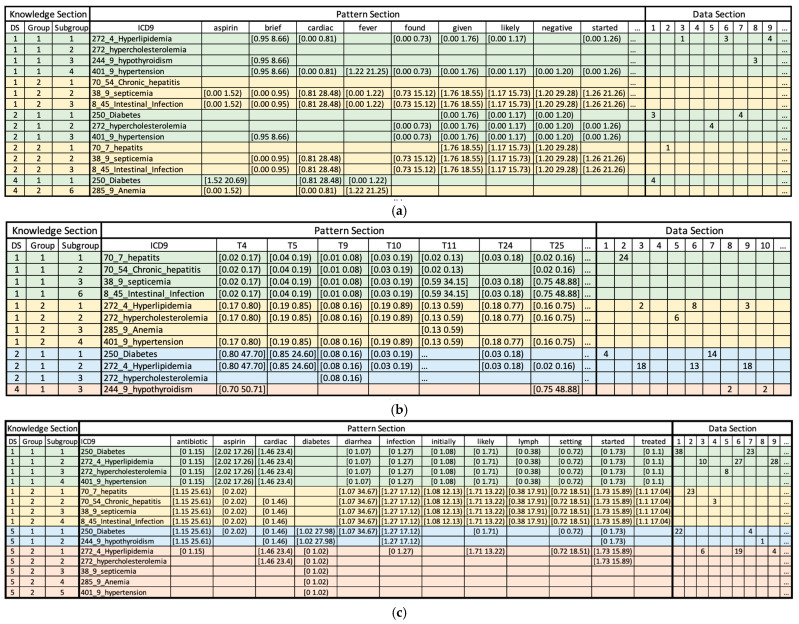
PDD Knowledge Base. (**a**) Output with the TF-IDF dataset; Rows highlighted in green and yellow represent two opposite pattern groups. (**b**) output with TM dataset; Rows highlighted in green, yellow, blue, and pink correspond to four distinct pattern groups: DSU[1,1,*], DSU[1,2,*], DSU[2,1,*], and DSU[4,1,*].(**c**) output with TF-IDF bigram-included dataset; Rows highlighted in green, yellow, blue, and pink correspond to four distinct pattern groups: DSU[1,1,*], DSU[1,2,*], DSU[5,1,*], and DSU[5,2,*]. As an example, the first row in (**a**) can be described as follows: The Knowledge Section shows the pattern is clustered within DSU[1,1,1], representing the first disentangled space, the first group, and the first subgroup. The Pattern Section indicates that the ICD9 code 272.4 is associated with the token ‘brief’, with a probability ranging from 0.95% to 8.66%, with ‘cardiac’ ranging from 0% to 0.81%, with ‘found’ ranging from 0% to 0.73%, and so on. The Data Section reveals that this pattern appears in record 3 once, in record 6 three times, and in record 9 four times.

**Figure 3 bioengineering-12-00308-f003:**
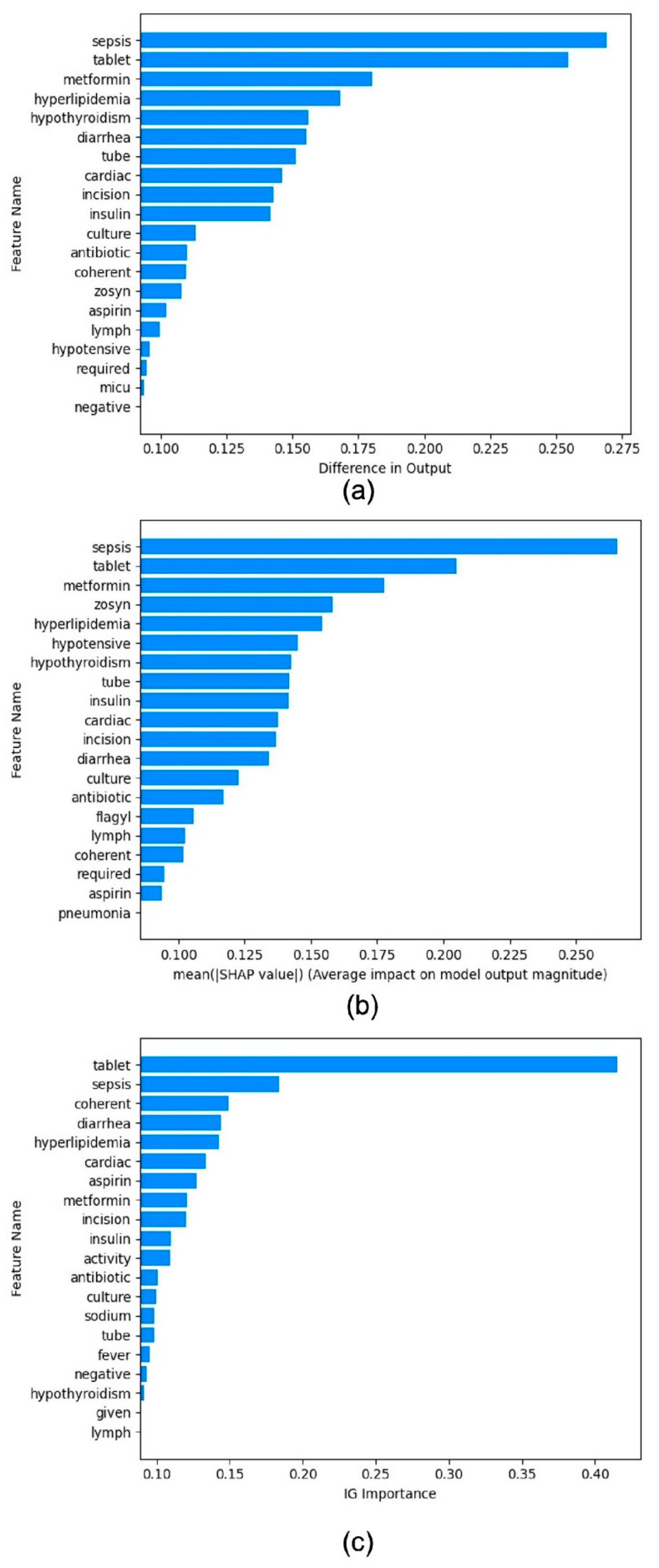
Mean feature importance of the top 20 predictors for unspecified septicemia (ICD-9 code: 38.9) across different methods. (**a**) Feature Permutation. (**b**) Gradient SHAP. The values represent the average impact on model output magnitude. (**c**) Integrated Gradient.

**Figure 4 bioengineering-12-00308-f004:**
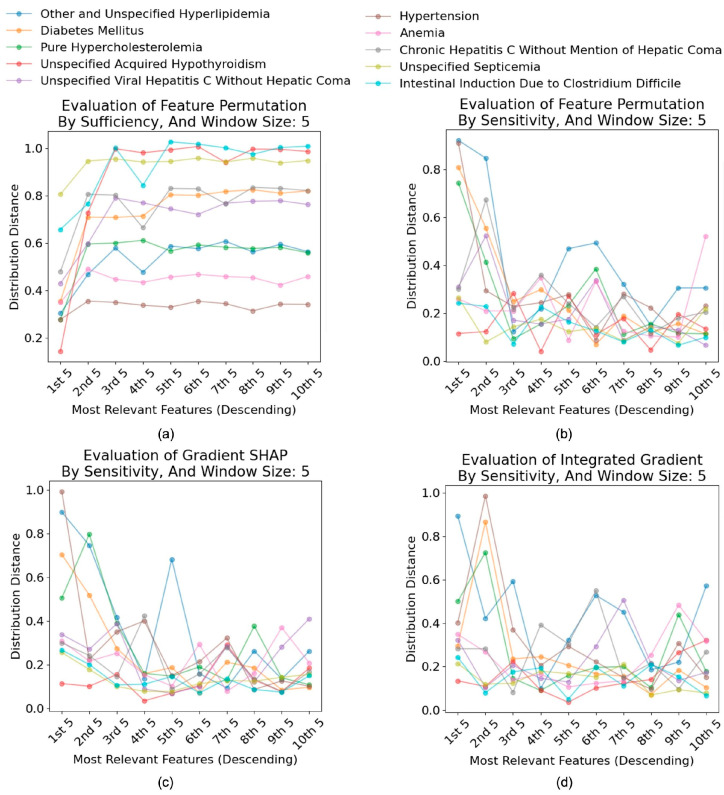
Evaluation of interpretability tools (Window Size: 5).

**Table 1 bioengineering-12-00308-t001:** Description of the top 10 ICD-9 codes in the MIMIC-IV dataset.

ICD-9 Code	Disease Name	Frequency	Distribution
250.0	Diabetes mellitus without mention of complication, type II or unspecified type, not stated as uncontrolled	12,539	24.1246%
272.4	Other and unspecified hyperlipidemia	11,840	22.7797%
244.9	Unspecified acquired hypothyroidism	10,224	19.6706%
272.0	Pure hypercholesterolemia	3411	6.5626%
38.9	Unspecified septicemia	2878	5.5372%
401.9	Unspecified essential hypertension	2521	4.8503%
70.54	Chronic hepatitis C without mention of hepatic coma	2466	4.7445%
8.45	Intestinal induction due to Clostridium difficile	2327	4.4771%
70.7	Unspecified viral hepatitis C without hepatic coma	2069	3.9807%
285.9	Anemia, unspecified	1701	3.2727%

**Table 2 bioengineering-12-00308-t002:** Benchmarking prediction models for the top 10 ICD-9 codes using TF-IDF and Topic Modeling (TM) features (the best-performing model in each metric is highlighted in bold).

Data	Model	Macro ROC-AUC	Micro ROC-AUC	Macro F1	Micro F1	Balanced Acc.	Macro Prec.	Micro Prec.	Macro Recall	Micro Recall
TM_Top10	CNN	0.784	0.842	0.262	0.360	0.274	0.288	0.360	0.274	0.360
	GRU	0.781	0.840	0.257	0.365	0.279	0.263	0.365	0.279	0.365
	LSTM	0.764	0.831	0.258	0.353	0.277	0.325	0.353	0.277	0.353
	MultiResCNN	**0.793**	**0.846**	0.269	**0.373**	0.289	0.267	**0.373**	0.289	**0.373**
	SVM	0.776	0.838	0.266	0.368	0.283	0.268	0.368	0.283	0.368
	RF	0.776	0.838	0.263	0.367	0.278	0.307	0.367	0.278	0.367
	PDD	0.689	0.643	**0.370**	0.358	**0.447**	**0.440**	0.358	**0.450**	0.358
TF-IDF_Top10	CNN	0.950	0.963	0.623	0.738	0.604	0.671	0.738	0.604	0.738
	GRU	0.941	0.958	0.543	0.713	0.523	0.604	0.713	0.523	0.713
	LSTM	0.548	0.721	0.036	0.223	0.100	0.036	0.223	0.100	0.223
	MultiResCNN	**0.956**	**0.968**	0.611	**0.743**	0.596	**0.689**	**0.743**	0.596	**0.743**
	SVM	0.947	0.962	0.514	0.692	0.509	0.658	0.692	0.509	0.692
	RF	0.947	0.962	0.58	0.731	0.558	0.662	0.731	0.558	0.731
	PDD	0.825	0.855	**0.640**	0.738	**0.679**	0.650	0.738	**0.680**	0.738
TF-IDF_Top10 (Bigram included)	CNN	0.943	0.960	0.616	0.736	0.603	0.647	0.736	0.603	0.736
	GRU	0.933	0.955	0.507	0.702	0.526	0.558	0.702	0.526	0.702
	LSTM	0.937	0.958	0.570	0.713	0.575	0.598	0.713	0.575	0.713
	MultiResCNN	**0.948**	**0.966**	0.598	0.739	0.589	0.652	0.739	0.589	0.739
	SVM	0.942	0.962	0.591	0.719	0.586	0.614	0.719	0.586	0.719
	RF	0.942	0.962	0.604	0.738	0.589	0.646	0.738	0.589	0.738
	PDD	0.909	0.890	**0.820**	**0.802**	**0.841**	**0.822**	**0.802**	**0.841**	**0.802**

**Table 3 bioengineering-12-00308-t003:** Evaluation of interpretability tools per ICD-9 code and its average by comprehensiveness, sufficiency, and sensitivity.

Interpretability Method	Faithfulness Criteria	ICD_9 Code										Average
		272.4	250.0	272	244.9	70.7	401.9	285.9	70.54	38.9	8.45	
Gradient SHAP	Comprehensiveness	0.391	0.596	0.489	1.00	0.390	0.194	0.269	0.435	0.474	0.820	0.506
	Sufficiency	0.1965	0.216	0.203	0.106	0.316	0.227	0.288	0.379	0.502	0.275	0.271
	Sensitivity	0.908	0.825	0.727	0.495	0.538	0.796	0.515	0.479	0.292	0.266	0.584
Feature Permutation	Comprehensiveness	0.404	0.591	0.503	0.999	0.409	0.205	0.288	0.467	0.468	0.783	0.510
	Sufficiency	0.192	0.212	0.206	0.102	0.310	0.220	0.281	0.344	0.518	0.290	0.267
	Sensitivity	0.928	0.842	0.748	0.475	0.514	0.761	0.559	0.528	0.287	0.265	0.591
TIIntegrated Gradient	Comprehensiveness	0.410	0.573	0.491	0.981	0.356	0.220	0.277	0.424	0.426	0.679	0.484
	Sufficiency	0.192	0.221	0.213	0.107	0.369	0.234	0.304	0.388	0.569	0.382	0.298
	Sensitivity	0.856	0.806	0.751	0.490	0.455	0.760	0.514	0.457	0.277	0.262	0.563

## Data Availability

Access to the MIMIC IV dataset can be requested at URL (accessed on 11 October 2024) https://physionet.org/content/mimiciv/0.4/.

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
