# Peer review of "Benchmarking Interpretability in Healthcare Using Pattern Discovery and Disentanglement"

_bioengineering, 2025, doi:10.3390/bioengineering12030308_

Round 1
Reviewer 1 Report
Comments and Suggestions for Authors
Here by use of the MIMIC-IV dataset, free-text clinical notes and ICD-9 codes with
Term Frequency-Inverse Document Frequency and Topic Modeling are processed. Here PDD algorithm discretizes numerical features into event-based features, and discovers association patterns from a disentangled statistical feature value association space, and clusters clinical records, its a good contribution.
Always introduction should be introducing the background and why we need this work and there should be a separate section named literature surveys where all the related past works should be mentioned and highlight their advantages and disadvantages and then conclude this section with problem statement and then proposed approach should come. So split the section 1 into two different sections.
Present section 2 should be section 3 names as proposed model. This is described systematic and its perfect.
Present section results and discussions have many remarks so make it all plural like Results and Discussions.
As said Section 4 should be section 2 now.
Exciting thing is PDD has achieved performance comparable to existing methods, validating its effectiveness in machine learning. That's so good.
Results have make it clear that findings reinforce the importance of integrating inherently interpretable models in clinical environments to ensure reliable and explainable AI-driven decision-making.
Now add future works in details in conclusions.
Author Response
Comments 1: Here by use of the MIMIC-IV dataset, free-text clinical notes and ICD-9 codes with
Term Frequency-Inverse Document Frequency and Topic Modeling are processed. Here PDD algorithm discretizes numerical features into event-based features, and discovers association patterns from a disentangled statistical feature value association space, and clusters clinical records, its a good contribution.
Response 1: We sincerely appreciate the reviewer’s positive feedback and recognition of our contributions. We are pleased that the reviewer found our approach valuable.
Comments 2: Always introduction should be introducing the background and why we need this work and there should be a separate section named literature surveys where all the related past works should be mentioned and highlight their advantages and disadvantages and then conclude this section with problem statement and then proposed approach should come. So split the section 1 into two different sections. Present section 2 should be section 3 names as proposed model. This is described systematic and its perfect. Present section results and discussions have many remarks so make it all plural like Results and Discussions. As said Section 4 should be section 2 now.
Response 2: We appreciate the reviewer’s suggestion regarding the structure of the paper. In response, we have reorganized the manuscript into five sections: (1) Introduction, (2) Related Work, (3) Proposed Model, (4) Results and Discussions, and (5) Conclusion. Additionally, we have revised all instances of 'result' and 'discussion' to their plural forms for consistency. We thank the reviewer for this valuable feedback.
Comments 3: Exciting thing is PDD has achieved performance comparable to existing methods, validating its effectiveness in machine learning. That's so good.
Results have make it clear that findings reinforce the importance of integrating inherently interpretable models in clinical environments to ensure reliable and explainable AI-driven decision-making.
Response 3: We sincerely appreciate the reviewer’s positive feedback and recognition of our contributions. We are pleased that the reviewer found our approach valuable.
Comments 4: Now add future works in details in conclusions.
Response 4: Thank you for pointing this out. We have added “future work” at the end of Conclusion section. We have highlighted all revisions in blue in the updated manuscript.
Reviewer 2 Report
Comments and Suggestions for Authors
The manuscript presents valuable insights into AI interpretability in healthcare, but certain aspects require refinement before acceptance in the current form. Below are some of the observations that need to be addressed:
- The study focuses on DL models but missing the comparison with ML models.
- The class imbalance for ICD-9 code is mentioned but not addressed if there was any over or under sampling is applied.
- The paper does not discuss the computational cost or scalability of the PDD method compared to post-hoc interpretability techniques.
- The references are not in order, please organize them.
- Caption of Fig 1 needs to be addressed properly. The current one doesn’t make sense for the figure. Within the figure there is a term “process data” used multiple times.
- Figure 2 seems like an image of the tables and is not clear, instead maybe make a table.
- Some references are missing, as in section 4.1 Horng et al. ref is not mentioned.
Author Response
Comments 1: The manuscript presents valuable insights into AI interpretability in healthcare, but certain aspects require refinement before acceptance in the current form. Below are some of the observations that need to be addressed:
Response 1: We sincerely appreciate the reviewer’s feedback. Below is our response.
Comments 2: The study focuses on DL models but missing the comparison with ML models.
Response 2: Thank you for your comment. We have added two machine learning models, Support Vector Machine (SVM) and Random Forest, to the comparison results across all evaluation metrics in Table 2. We have highlighted all revisions in blue in the section “results and discussions” in the updated manuscript.
Comments 3: The class imbalance for ICD-9 code is mentioned but not addressed if there was any over or under sampling is applied.
Response 3: Thank you for pointing this out. As mentioned in our previous work, the PDD algorithm is designed to handle imbalanced data effectively. To ensure consistency in our experimental setup, we chose not to apply any oversampling or undersampling techniques to the deep learning (DL) or machine learning (ML) models. We appreciate the opportunity to clarify this point. We have add our discussion in the section 4.1 in the revised manuscript.
Comments 4: The paper does not discuss the computational cost or scalability of the PDD method compared to post-hoc interpretability techniques.
Response 4: We appreciate the reviewer’s insightful comment. To address this, we have added Section 4.5: Computational Cost Discussion. The general idea is that post-hoc interpretability techniques require additional computational overhead for an already complex model. In contrast, the PDD algorithm is inherently interpretable, eliminating the need for additional computations and making it a more scalable alternative. We thank the reviewer for this valuable suggestion, which has strengthened our discussion.
Comments 5: The references are not in order, please organize them.
Response 5: Thanks for pointing out this. We have checked all references and re-organized them.
Comments 6: Caption of Fig 1 needs to be addressed properly. The current one doesn’t make sense for the figure. Within the figure there is a term “process data” used multiple times.
Response 6: Thank you for pointing this out. We have revised Figure 1 and updated the caption to "Figure 1. Overview of the benchmarking process: Comparison of benchmarking models and the PDD for interpretability." The revisions have been highlighted in the manuscript. We appreciate your feedback.
Comments 7: Figure 2 seems like an image of the tables and is not clear, instead, maybe make a table.
Response 7: Thank you for your thoughtful comment. We sincerely appreciate your suggestion and attempted to convert the figure into a table. However, the journal's formatting guidelines prohibit vertical lines, making it challenging to clearly separate the knowledge, pattern, and data sections as intended for the Knowledge Base. Additionally, the differently colored cells (yellow, green, red) convey important meanings, representing different groups of diseases (ICD-9 codes). Unfortunately, these distinctions would be difficult to maintain in a standard table format. Therefore, we have decided to retain Figure 2 as three separate figures. We will also upload high-resolution versions in the final submission. We sincerely appreciate your feedback and are happy to consider any further suggestions.
Comments 8: Some references are missing, as in section 4.1 Horng et al. ref is not mentioned.
Response 8: Thanks for pointing out this. We have checked all references and re-organized them, especially the Horng et al. ref in section 2.1. We have highlighted all revisions in blue in the updated manuscript.
Round 2
Reviewer 1 Report
Comments and Suggestions for Authors
Now all the necessary changes are done, its a good contribution.